# Impact of Oncogenic Changes in p53 and KRAS on Macropinocytosis and Ferroptosis in Colon Cancer Cells and Anticancer Efficacy of Niclosamide with Differential Effects on These Two Processes

**DOI:** 10.3390/cells13110951

**Published:** 2024-05-30

**Authors:** Nhi T. Nguyen, Souad R. Sennoune, Gunadharini Dharmalingam-Nandagopal, Sathish Sivaprakasam, Yangzom D. Bhutia, Vadivel Ganapathy

**Affiliations:** Department of Cell Biology and Biochemistry, Texas Tech University Health Sciences Center, Lubbock, TX 79430, USA; nhi.t.nguyen@ttuhsc.edu (N.T.N.); souad.sennoune@ttuhsc.edu (S.R.S.); gnandago@ttuhsc.edu (G.D.-N.); sathish.sivaprakasam@ttuhsc.edu (S.S.); yangzom.d.bhutia@ttuhsc.edu (Y.D.B.)

**Keywords:** p53 deletion, KRAS mutation, SLC7A11, SLC38A5, ferroptosis, macropinocytosis, antioxidant machinery, lipid peroxidation, niclosamide, colon cancer

## Abstract

Mutations in p53 and KRAS are seen in most cases of colon cancer. The impact of these mutations on signaling pathways related to cancer growth has been studied in depth, but relatively less is known on their effects on amino acid transporters in cancer cells. This represents a significant knowledge gap because amino acid nutrition in cancer cells profoundly influences macropinocytosis and ferroptosis, two processes with opposing effects on tumor growth. Here, we used isogenic colon cancer cell lines to investigate the effects of p53 deletion and KRAS activation on two amino acid transporters relevant to macropinocytosis (SLC38A5) and ferroptosis (SLC7A11). Our studies show that the predominant effect of p53 deletion is to induce SLC7A11 with the resultant potentiation of antioxidant machinery and protection of cancer cells from ferroptosis, whereas KRAS activation induces not only SLC7A11 but also SLC38A5, thus offering protection from ferroptosis as well as improving amino acid nutrition in cancer cells via accelerated macropinocytosis. Niclosamide, an FDA-approved anti-helminthic, blocks the functions of SLC7A11 and SLC38A5, thus inducing ferroptosis and suppressing macropinocytosis, with the resultant effective reversal of tumor-promoting actions of oncogenic changes in p53 and KRAS. These findings underscore the potential of this drug in colon cancer treatment.

## 1. Introduction

As per the most recent colorectal cancer statistics of 2023, colorectal cancer is the third most commonly diagnosed cancer in men and women in the United States [1,2,3]. It is also the cause of the second most cancer-related deaths in men and the most prominent cause of cancer-related deaths in younger men (<50 years of age). Colorectal cancer is the cause of approximately one million deaths worldwide every year. There are several risk factors for colorectal cancer, including age, sex, family history, diet, inflammation, and gene mutations. Truly alarming is the estimate that about 4% of men and women will be diagnosed with colorectal cancer at some point in their lifetime [2]. Due to increased implementation of routine screening procedures and resultant early detection and also due to newly developed drugs for effective treatment, the 5-year survival in patients with colorectal cancer has been steadily increasing over the years, from about 50% in 1975 to about 70% in 2015 in most countries, including the United States [4].

Several germline and somatic mutations are associated with colorectal cancer [5,6,7]. These mutations occur in tumor suppressor genes, oncogenes, and DNA repair genes. A small percentage of colorectal cancer is inheritable, where germline mutations are found in the genes coding for the adenomatous polyposis coli (APC) (Gardner’s syndrome), the tumor suppressor LKB1 (Peutz–Jeghers syndrome), DNA repair proteins MLH1, MSH2, MSH6, PMS2, EPCAM, and MUTYH (Lynch’s syndrome), and cystic fibrosis transmembrane conductance regulator CFTR (cystic fibrosis) [5,6,7]. Most cases of colorectal cancer are sporadic, and somatic mutations underlie carcinogenesis in these cases. Most often, mutations in APC with the resultant activation of β-catenin-dependent signaling and transcription initiate rapid cell proliferation, followed by mutations in the oncogene KRAS and subsequently in the tumor suppressor p53 that mediate and maintain cellular transformation and cancer growth [8,9,10]. APC is an integral part of the destruction complex that promotes proteasomal degradation of β-catenin; therefore, loss-of-function mutations in APC, which prevent β-catenin degradation, are associated with colorectal cancer. Mutations in KRAS related to colorectal cancer result in gain-of-function with a decrease in GTPase activity of RAS and hence an increase in GTP-bound RAS and consequent persistent activation of the signaling pathways responsible for cell proliferation and carcinogenic cellular transformation. Mutations in p53 found in colorectal cancer interfere with the transcriptional activity of the protein, thus protecting the cells from apoptosis and promoting cell proliferation and growth.

Macropinocytosis and ferroptosis are two biological phenomena that have come to the forefront in the field of cancer biology in relatively more recent years. Both of these processes are connected to cancer cell nutrition: macropinocytosis to amino acid nutrition [11,12] and ferroptosis to iron/amino acid nutrition [13,14]. Oncogenic mutations in KRAS are associated with the induction of macropinocytosis that mediates the uptake of extracellular proteins for subsequent hydrolysis when macropinosomes fuse with lysosomes followed by the use of the resultant amino acids in cellular metabolism. Tumor-associated blood vessels are often leaky [15,16], thus facilitating the release of plasma proteins such as albumin into the extracellular milieu that are used in macropinocytosis. Even though the incidence of KRAS mutations is significant in colorectal cancer (30–50%), it occurs far more frequently in pancreatic cancer (~90%) [17]. As such, the association between KRAS mutations and macropinocytosis has been investigated mostly in relation to pancreatic cancer [11,12]. Macropinocytosis offers a novel mechanism for the acquisition of amino acids in cancer cells in addition to the import of extracellular amino acids via multiple amino acid transporters that are upregulated in cancer [18,19,20]. As such, macropinocytosis promotes cancer by supplying amino acids to support cell proliferation. Cancer cells are also obligatorily dependent on iron for growth and proliferation [21,22], but excess iron increases the risk for iron-induced cell death ferroptosis. Cancer cells find ways to evade this form of cell death by potentiating glutathione-dependent antioxidant machinery. Loss-of-function mutations in p53 play an important role in this process, where loss of p53 function leads to the induction of the cystine transporter SLC7A11 to promote the synthesis of the antioxidant peptide glutathione in cancer cells [23,24]. In the current study, we focused on the relative importance of an oncogenic mutation in KRAS (G12D) and deletion of p53 in macropinocytosis and ferroptosis in colon cancer using isogenic cancer cell lines with and without the KRAS mutation as well as with and without p53.

Current treatment options for colorectal cancer include removal of the cancerous tissue (surgery, radiofrequency ablation, and cryosurgery), radiation therapy, chemotherapy, and immunotherapy [25]. Monoclonal antibodies that neutralize the functions of VEGFRs and EGFRs on tumor cells and CTLA4 and PD-1 on cytotoxic T cells are increasingly used. Chemotherapy includes the use of 5-fluorouracil and 5-trifluoro-2-deoxythymidine (inhibitors of thymidylate synthetase), irinotecan (an inhibitor of topoisomerase I), oxaliplatin (a DNA-alkylating agent that interferes with DNA replication and transcription), and several small-molecule inhibitors of the tyrosine kinase activity associated with VEGFRs and EGFRs [26]. However, none of these therapies specifically target macropinocytosis and/or ferroptosis in cancer cells. In the present study, we identified niclosamide, an FDA-approved anti-helminthic drug [27,28], as not only a potent inhibitor of macropinocytosis but also an inducer of ferroptosis in colon cancer cells, thus underscoring the potential of this drug as an anticancer agent, especially for cancers associated with oncogenic changes in KRAS and p53.

## 2. Materials and Methods

### 2.1. Materials

[2,3-^3^H]-L-Serine (sp. radioactivity, 15 Ci/mmol) was procured from Moravek, Inc. (Brea, CA, USA). [^3^H]-Glutamate (sp. radioactivity, 50.8 Ci/mmol) was procured from PerkinElmer Corp (Waltham, MA, USA). Niclosamide was purchased from Millipore-Sigma (St. Louis, MO, USA), and ferrostatin-1 was obtained from Santa Cruz Biotechnology Inc. (Dallas, TX, USA). All other chemicals were obtained from Millipore-Sigma (St. Louis, MO, USA). The p53 activator RITA (reactivation of p53 and induction of tumor cell apoptosis) was obtained from Abcam (cat. No. ab219379) (Cambridge, UK).

### 2.2. Cell Lines and Culture Conditions

We used two pairs of isogenic human colon cancer cell lines: HCT-116/*p53*^+/+^ and HCT-116/*p53*^−/−^ cells (kindly provided by Dr. Bert Vogelstein, Johns Hopkins University School of Medicine, Baltimore, MD, USA). These are isogenic cell lines only with differences in terms of the presence or absence of p53. Both alleles of the *p53* gene were deleted in the parent HCT-116/*p53^+/+^* cell line by homologous recombination [29,30]. SW48 parent and KRAS*^G12D^* mutant SW48 cells were purchased from Horizon Discovery Ltd., Cambridge, UK. Additional relevant genotype features of these cell lines are given in the Section 3. SW48 cells were cultured in RPMI 1640 medium, supplemented with 2 mM L-glutamine and 25 mM sodium bicarbonate. HCT-116 cells were cultured in McCoY’s 5a medium. All media contained 10% fetal bovine serum. Cell cultures were tested every month for mycoplasma using a commercially available detection kit (cat. no. G238; Applied Biological Materials, Inc. Richmond, BC, Canada). All cell lines used in the present study were mycoplasma-free.

### 2.3. Uptake Measurement

Uptake of [^3^H]-serine was measured to monitor the transport activity of SLC38A5. SLC38A5 is Na^+^-coupled with H^+^ movement in the opposite direction; as such, the uptake was measured at pH 8.5 to generate an outward-directed H^+^ gradient. There are several Na^+^-coupled transporters for serine; therefore, we cannot specifically monitor the function of SLC38A5 by using a Na^+^ buffer. However, unlike other Na^+^-coupled transporters, SLC38A5 is functional when Na^+^ is replaced with Li^+^. Therefore, we replaced NaCl with LiCl in the uptake buffer. The uptake buffer was made up of 25 mM Tris/Hepes, pH 8.5, with 140 mM LiCl, 5.4 mM KCl, 1.8 mM CaCl_2_, 0.8 mM MgSO_4_, and 5 mM glucose. Serine is also transported by SLC7A5, a Na^+^-independent transporter; therefore, uptake via this transporter contributes to the total uptake measured in the Li^+^ buffer. Therefore, we included 5 mM tryptophan in the uptake buffer to compete with and block SLC7A5-mediated serine uptake; tryptophan is not a substrate for SLC38A5, and therefore, SLC38A5-mediated serine uptake is not affected by tryptophan. To determine the diffusional component in the total uptake of serine, the same uptake buffer but with LiCl replaced iso-osmotically with *N*-methyl-D-glucamine chloride (NMDGCl) was used. The uptake was measured in two buffers: (i) LiCl-buffer, pH 8.5 with 5 mM tryptophan; and (ii) NMDGCl-buffer, pH 8.5 with 5 mM tryptophan. The uptake in the NMDGCl buffer was subtracted from the uptake in the LiCl buffer to determine the transport activity of SLC38A5 [31].

SLC7A11 is a Na^+^-independent transporter; it mediates the entry of cystine into cells in exchange for intracellular glutamate. Notwithstanding this physiological mode of transport, we routinely measure its transport function by monitoring the uptake of radiolabeled glutamate. Under these conditions, SLC7A11 mediates the entry of radiolabeled glutamate into cells in exchange for intracellular unlabeled glutamate. Transport activity was measured using a Na^+^-free buffer (25 mM Hepes/Tris, 140 mM *N*-methyl-D-glucamine chloride, 5.4 mM KCl, 1.8 mM CaCl_2_, 0.8 mM MgSO_4_, and 5 mM glucose, pH 7.5). The diffusional component was calculated by measuring the uptake of [^3^H]-glutamate in the presence of excess unlabeled glutamate (5 mM). The transport activity of SLC7A11 was determined by subtracting the diffusional component from total uptake [32].

Cells were seeded in 24-well culture plates (2 × 10^5^ cells/well) and grown to reach confluence (2 or 3 days of culture depending on the cell line). On the day of uptake measurement, the culture plates were kept in a water bath at 37 °C. The medium was aspirated, and the cells were washed with the respective uptake buffer. The uptake medium (250 μL) containing a corresponding amino acid as the tracer was added to the cells. Following incubation for 15 min (SLC38A5) or 30 min (SLC7A11), the medium was removed, and the cells were washed thrice with ice-cold uptake buffer. The cells were then lysed in 1% sodium dodecyl sulfate/0.2 N NaOH and used for measurement of radioactivity.

### 2.4. Intracellular pH Measurement

The method used to monitor intracellular pH has been described in our previous report [31]. Cells were grown on rectangular coverslips (9 × 22 mm) until they reached confluence. The cells were then incubated with 7.5 μM SNARF-1-AM in a perifusion buffer at an extracellular pH of 7.4 for 30 min at 37 °C followed by a 30-min incubation in the same buffer but without the dye. This facilitated the hydrolysis of the SNARF-1-AM ester inside the cells. Two coverslips were placed back-to-back in a holder and perifused at a rate of 3 mL/min, and the fluorescence of SNARF-1 was monitored with a SLC-8100/DMX spectrofluorometer (Spectronics Instruments, Rochester, NY, USA). In situ calibration curves were generated as described previously [31]. Briefly, the cells on coverslips were perifused with a high-K^+^ buffer (10 mM NaCl, 146 mM KCl, 10 mM HEPES, 10 mM MES, 10 mM Bicine, 2 μM valinomycin, 6.8 μM nigericin, and 5 mM glucose, pH 5.5–8.0 at 0.2 pH intervals). The 644 nm/584 nm ratio of fluorescence was determined at each pH and then converted to pH using a modified Henderson–Hasselbalch equation. This calibration was used to calculate intracellular pH.

### 2.5. Macropinocytosis Assay

We used TMR (tetramethylrhodamine)-dextran (molecular weight, 70 kDa) to monitor macropinocytosis, as carried out by others [33,34] and described in our previous publication [31]. Briefly, cells were plated on coverslips, placed in wells in a 12-well plate at a density of 1 × 10^5^ cells/well, and cultured at 37 °C using the culture media appropriate for the given cell line. The cells were allowed to reach ~70% confluency. The cells were washed with a buffer consisting of 140 mM NaCl, 5.4 mM KCl, 1.8 mM CaCl_2_, 0.8 mM MgSO_4_, and 5 mM glucose, pH 7.5. The cells were then exposed to TMR-dextran (100 μg/mL) in the same buffer at 37 °C. Then, the cells were washed again with the respective buffer and then fixed with 4% paraformaldehyde for 5 min, washed several times with phosphate-buffered saline, and mounted using Prolong diamond with 4,6-diamidino-2-phenylindole (DAPI) as a nuclear marker. Cell images were taken using a Nikon T1-E microscope with an A1 confocal super-resolution module (Nikon, Dallas, TX, USA) with a 60× objective. The fluorescence quantification was performed by measuring the corrected total cell fluorescence (CTCF) using Image J (version: 2.14.0/1.54f) and the following formula:CTCF = (integrated density) − (area of cell of interest) × (mean fluorescence of background).

For groups of cells (15–20 cells/field) in an image, an outline was drawn to measure the integrated density, area of the cells of interest, and mean fluorescence of the adjacent background around the cells of interest. We randomly selected five separate fields with 15–20 cells and calculated the CTCF for each field; the five CTCF values were averaged.

### 2.6. RT-PCR

Total RNA was extracted from the cells using TRIzol Reagent (Thermo Fisher Scientific, Waltham, MA, USA), and the RNA was reverse-transcribed using a high-capacity cDNA reverse transcription kit (Thermo Fisher Scientific, Waltham, MA, USA). Quantitative PCR was performed with Power SYBR Green PCR master mix (Bio-Rad, Hercules, CA, USA). Primer sequences are given in Appendix A. The relative mRNA expression was determined by the 2^−∆∆Ct^ method with 18S mRNA for normalization.

### 2.7. Protein Isolation and Western Blot

The cells were lysed in Pierce™ RIPA buffer (Thermo Fisher Scientific, Waltham, MA, USA) supplemented with Halt™ Protease and Phosphatase Inhibitor Cocktail (Thermo Fisher Scientific, Waltham, MA, USA). Protein was measured using a Pierce™ BCA Protein Assay Kit (Thermo Fisher Scientific, Waltham, MA, USA). The samples, prepared in Laemmli Sample Buffer (Bio-Rad Labs, Hercules, CA, USA), were loaded onto SDS–PAGE gel and transferred onto a PVDF membrane. The membrane was incubated with antibodies diluted in 5% nonfat dry milk. Protein bands were visualized using Pierce™ ECL Western Blotting Substrate (Thermo Fisher Scientific, Waltham, MA, USA) and developed on an autoradiography film. Most antibodies were purchased from Cell Signaling Technology (Danvers, MA, USA): FTH (#4393), GPX4 (#52455), HSP60 (#12165), SLC7A11 (#12691), and SLC3A2 (#47213). Antibodies for p53 (#SC-126) and Myc (#SC-40) were obtained from Santacruz Biotechnology (Dallas, TX, USA). Secondary antibody horseradish-peroxidase-conjugated goat anti-rabbit IgG (#1706515) was obtained from Bio-Rad Labs (Hercules, CA, USA). For quantification of protein levels by densitometry, the experiment was carried out in triplicate.

### 2.8. Assays for Lipid Radicals (Ferroptosis) and Iron

A lipid radical (ferroptosis) assay and iron assay were carried out as originally described by others [35,36] and used in one of our previous publications [30]. Cells were cultured on a 25 mm glass coverslip to ~70% confluency. The cells were washed with NaCl buffer, pH 7.5, and then incubated with 1 µM of LipiRadical Green (FDV-0042, Funakoshi, Tokyo, Japan) or Ferro-orange (F374, Dojindo, Rockville, MD, USA) in NaCl buffer, pH 7.5, for 20 min and then washed. To analyze the effect of niclosamide on lipid peroxidation, the cells were treated with a fluorescence probe along with niclosamide for 20 min and then washed. The glass coverslip containing the cells was then probed under an inverted microscope. The fluorescence imaging was captured using a Nikon T1-E microscope with an A1 confocal super-resolution module (Nikon, Dallas, TX, USA) with a 60× objective. For the ferroptosis assay, the excitation was at 470 nm and the emission was at 520 nm; for the iron assay, the excitation was at 560 nm and the emission was at 620 nm. The images represent a maximum projection intensity derived from a Z-stack. The fluorescence quantification was performed by measuring the corrected total cell fluorescence (CTCF) using Image J (version: 2.14.0/1.54f) and the following formula: CTCF = (integrated density) − (area of cell of interest) × (mean fluorescence of background).

### 2.9. Glutathione and Lipid Peroxidation Assay

Cells cultured under various experimental conditions were used to measure glutathione (GSH-Glo assay, Promega, Madison, WI, USA). Malondialdehyde (MDA) was quantified using a lipid peroxidation kit (MAK085, Millipore-Sigma, St. Louis, MO, USA).

### 2.10. Colony Formation Assay

A colony formation assay was carried out with different doses of niclosamide. The initial seeding was 500 cells/well, and the culture was continued for 10 days with the culture medium replaced with fresh medium on alternate days. At the end of the 10-day period, the medium was removed, and the colonies were fixed with ice-cold methanol/acetone and then stained with Giemsa stain. After examination, lysis buffer was added (1% sodium dodecyl sulfate/0.2 N NaOH), and the extracted chromophore was quantified using a Microplate Reader (Glomax multi-detection system, Promega, Madison, WI, USA).

### 2.11. MTT Assay

An MTT assay was used to assess metabolic activity in the cells, a substitute for cell vaibility and proliferation. This involved the NAD(P)H-dependent reduction of the tetrazolium dye MTT, which is 3-(4,5-dimethylthiazol-2-yl)-2,5-diphenyltetrazolium bromide. Cells were seeded in 96-well plates and cultured for 24 h, after which the cells were exposed to niclosamide. The cells were cultured for an additional 72 h with fresh medium supplied every 24 h. The cells were then washed twice with phosphate-buffered saline followed by MTT reagent (ATCC, Manassas, VA, USA). Treatment and lysis of the cells were carried out as per the manufacturer’s instructions. Absorbance of the lysate was measured at 550 nm.

### 2.12. Chromatin-Immunoprecipitation (ChIP) Assay

A ChIP assay was carried out using an EZ-Magna ChIP A/G kit (Millipore, Burlington, MA, USA). SW48 cells were treated with 1% formaldehyde for 10 min to crosslink proteins and nucleic acids. The contents were collected in phosphate-buffered saline, supplemented with Halt™ Protease and Phosphatase Inhibitor Cocktail (Thermo Fisher Scientific, Waltham, MA, USA), and lysed in a nuclear lysis buffer. The lysate was sonicated using BioRuptor Plus (Diagenode, Denville, NJ, USA) to shear DNA into fragments of 200–1000 base-pairs. DNA concentration was measured using a NanoDrop Spectrophotometer (Thermo Fisher Scientific, Waltham, MA, USA), and 100 μg of DNA was used for immunoprecipitation with anti-p53 (#SC-126) and anti-Myc (#SC-40) antibodies (Santacruz Biotechnology, Dallas, TX, USA) or mouse IgG (Millipore-Sigma, St. Louis, MO, USA). Before immunoprecipitation, an aliquot of the supernatant was removed for use as an input. Immunoprecipitated DNA was isolated on the column, and the relative enrichment of p53 and Myc on SLC38A5 promoter was assessed by PCR (primer sequences are given in Appendix A). In some experiments, SW48 cells were treated with the p53 activator RITA (1 μM) for 16 h prior to the ChIP assay.

### 2.13. Assay for Reactive Oxygen Species (ROS) with 2′,7′-Dichlorodihydrofluorescein Diacetate Staining

A fluorecence probe composed of DCFH-DA was used to measure ROS [37]. Briefly, cells were grown in a 96-well plate and then incubated with DCFH-DA (10 µM) at 37 °C for 30 min in the dark. At the end of the incubation, the cells were treated with different concentrations of niclosamide. Fluorescence intensity was monitored with a Microplate Reader (Glomax multi-detection system, Promega, Madison, WI, USA) at the excitation and emission wavelengths of 485 and 528 nm, respectively. Cellular fluorescence levels were expressed as per ug of protein.

### 2.14. Statistics

Uptake experiments were carried out in triplicates, and each experiment was repeated thrice using independent cell cultures. Statistical analysis was conducted with Student’s two-tailed, paired *t*-test for single comparison, and a *p*-value < 0.05 was taken as evidence of statistical significance. Data are provided as means ± S.E. For quantification of fluorescence signals in macropinocytosis and ferroptosis, ANOVA followed by Dunn’s test was used to determine the significance of differences among different groups.

## 3. Results

### 3.1. Origin and Genetic Background of Colon Cancer Cell Lines Used in This Study

Two pairs of isogenic human colon cancer cell lines were used in this study. The parent HCT-116 cell line was originally derived from colorectal adenocarcinoma in a 48-year-old male. This cell line possesses wild-type APC and wild-type p53 but mutant β-catenin where the mutation stabilizes the protein, thus leading to persistent transcriptional activity of β-catenin. KRAS in this cell line harbors the oncogenic mutation G13D. PTEN and BRAF are wild-type in this cell line [38,39,40,41,42]. This cell line was used to generate a p53-null cell line in the laboratory of Dr. Vogelstein (Johns Hopkins University School of Medicine, Baltimore, MD, USA). The parent SW48 cell line was originally derived from colorectal adenocarcinoma in an 83-year-old female. This cell line possesses wild-type APC, KRAS, p53, PTEN, and BRAF but mutant β-catenin that stabilizes the protein with resultant persistent transcriptional activity [38,39,40,41,42]. This cell line was used to generate a KRAS mutant (G12D) cell line by Horizon Discovery Ltd. (Cambridge, UK). The status of APC, KRAS, and p53 in these two pairs of isogenic cell lines, which are directly relevant to the present study, are given in Table 1.

As such, when macropinocytosis and ferroptosis and their associated cellular pathways are compared between the isogenic cell lines HCT-116 and HCT-116/p53 KO, the observed differences are most likely due to the presence or absence of functional p53 in a colon cancer cell line on the background of an oncogenic KRAS mutation and an active β-catenin signaling pathway. Similarly, when the isogenic cell lines SW48 and SW48/KRAS mutant are used for comparison, the observed differences are most likely due to the presence or absence of an oncogenic mutation in KRAS in a colon cancer cell line on the background of functional p53 and an active β-catenin signaling pathway.

### 3.2. Impact of p53 Deletion and KRAS-G12D Mutation on SLC38A5 Expression and Function

Several studies have demonstrated the involvement of Na^+^/H^+^ exchanger in macropinocytosis [43,44]. Here, the exchanger-mediated alkalinization of the cytoplasmic domain of the plasma membrane influences sub-membranous the reorganization of the actin cytoskeleton to form membrane ruffles that promote invagination and closure of the plasma membrane to form macropinosomes, a non-clathrin-/non-receptor-mediated endocytic process. Recently, we found that the amino acid transporter SLC38A5, which functions as an amino-acid-dependent Na^+^/H^+^ exchanger, is also able to induce macropinocytosis in breast cancer cells [45]. SLC38A3, another amino acid transporter that functions similarly to SLC38A5, may also affect macropinocytosis [46], but this has not yet been investigated experimentally. In the present study, we examined the influence of p53 and KRAS mutation on the expression and function of SLC38A3 and SLC38A5 by RT-PCR and transport function. The deletion of *p53* in HCT-116 cells decreased the transport function of SLC38A3/SLC38A5 significantly even though the steady-state levels of SLC38A3 mRNA and SLC38A5 mRNA appeared to be little affected (Figure 1A,B). In the SW48 cells, the presence of the oncogenic mutation in KRAS (G12D) induced the expression (i.e., increase in mRNA levels) and function (i.e., transport activity) of both transporters (Figure 1A,B). The changes observed in Na^+^-coupled serine uptake may not have been entirely due to changes in SLC38A3/SLC38A5 expression, but the changes in Li^+^-coupled serine uptake certainly were. Even though we were not able to differentiate between SLC38A3 and SLC38A5 in terms of Li^+^-coupled serine uptake (no selective inhibitors available yet) in these cell lines, we believe that SLC38A5 was the predominant contributor to the observed uptake based on the relative levels of the respective mRNAs, as determined from ΔCt values in qRT-PCR (ratio of SLC38A5: SLC38A3 mRNA level was 8:1 in the HCT-116 cells and 52:1 in the SW48 cells). Based on these data, we conclude that even though p53 is a tumor suppressor whereas KRAS-G12D is an oncogenic mutation, both induce SLC38A5 expression.

### 3.3. Interaction of p53 and Myc with SLC38A5 Promoter

The function of p53 as a transcription factor is well known. In contrast, the downstream pathway for RAS involves phosphorylation cascades [47,48]. The gene expression induced by RAS signaling is mediated by several transcription factors, including Ets, Fos, Jun, and Myc [49,50]. Here, we focused on Myc because of its role as an oncogene and also because of a previous report showing that SLC38A5 (identified as SN2 in the report) is induced in cancer cells upon the ectopic expression of Myc [51]. Since our experiments showed that both p53 and the KRAS mutant induced SLC38A5 expression, we performed ChIP to demonstrate the binding of p53 and Myc to SLC38A5 gene promoter. First, we analyzed the promoter sequence for the p53- and Myc-binding sites using the eukaryotic promoter database (https://epd.expasy.org/epd) (accessed on 10 January 2024), a webtool created and managed by the Computational Cancer Genomics lab of the Swiss Institute of Bioinformatics. We found numerous binding motifs for both transcription factors in human *SLC38A5* gene promoter (Figure 2A). The binding sites were confirmed by the publicly available JASPAR CORE 22 database (https://genome.ucsc.edu) (accessed on 2 May 2024). We then monitored the protein levels for p53 and Myc in all four cell lines used in the study (Figure 2B). This confirmed the absence of p53 in the HCT-116/p53 KO cells. More importantly, the Myc protein levels were higher in the SW48/KRAS mutant cells than in the parent SW48 cells. Interestingly, the Myc levels were also higher in the *p53*-null HCT-116 cells than in the parent HCT-116 cells. The ChIP experiment demonstrated the binding of both p53 and Myc to the *SLC38A5* promoter (Figure 2C). The regions of the promoter covered by the PCR primers in this experiment were as follows: −936 to −1139 for Myc; −4027 to −4241 for p53. The binding of p53 to the promoter was further confirmed by treating the SW48 cells with the p53 activator (RITA; 1 μM; 16 h treatment) (Figure 2C,D). This treatment stabilized the p53 protein and increased the steady-state levels of p53 in the colon cancer cells; the observed increase was the highest for the parent SW48 cells that were used for the ChIP assay (Figure 2E). To further confirm the binding of p53 to the *SLC38A5* gene promoter, we used a negative control with a PCR primer pair that covered the region from –1841 to –2064 in the promoter where there were no predicted binding sites for p53. This primer pair failed to yield a PCR product in the ChIP assay with or without the treatment with RITA (Figure 2C). The observed increase in the expression of SLC38A5 in the SW48/KRAS mutant cells agrees with the increased levels of Myc in these cells. In contrast, the observed decrease in the expression of SLC38A5 in the *p53*-null HCT-116 cells in spite of the increase in Myc protein levels suggests a more predominant role for p53 than for Myc in the control of SLC38A5 expression in this particular cell line.

### 3.4. Impact of p53 Deletion and KRAS Mutation on Macropinocytosis

We monitored macropinocytosis using TMR-dextran in all four cell lines to determine how the deletion of p53 in HCT-116 cells and the presence of the oncogenic KRAS mutation G12D in SW48 cells affect this process (Figure 3). Macropinocytosis was robust in the HCT-116 cells that already harbored the oncogenic mutation G13D in KRAS; deletion of p53 in this cell did not have any noticeable effect on macropinocytosis. In contrast, macropinocytosis was relatively lower in the SW48 parent cells than in the HCT-116 parent cells, and the presence of the oncogenic mutation G12D in KRAS in the SW48 cells markedly increased macropinocytosis. Collectively, these findings show that the process of macropinocytosis in colon cancer cells is controlled by KRAS, with little impact from p53.

### 3.5. Influence of p53 Deletion and KRAS Mutation on the Expression and Function of SLC7A11

The primary mechanism for the protection of cells from iron-induced cell death, ferroptosis, is glutathione, which removes lipid peroxides by the activity of glutathione peroxidase [52]. Accordingly, the amino acid transporter SLC7A11, which provides the limiting amino acid cysteine for glutathione synthesis, is an important controller of ferroptosis in cancer cells [53,54]. The recruitment of SLC7A11 to the plasma membrane requires a protein chaperone, called SLC3A2, and the functionally active complex is a heterodimer consisting of the chaperone SLC3A2 and the actual transporter SLC7A11 [55,56]. Therefore, we examined the expression of the transporter as well as the chaperone in the four cancer cell lines. We found the expression of both components of the transport system to be increased markedly by p53 deletion and KRAS-G12D mutation (Figure 4A,B). This increase in mRNA levels corresponded to an increase in transport function, as monitored by glutamate uptake (Figure 4C,D) and protein levels (Figure 4E,F). These data show that p53 is a suppressor of SLC7A11/SLC3A2 expression, whereas the oncogenic mutation G12D in KRAS is an inducer. This is in contrast to the effects of p53 and KRAS mutation on SLC38A5 expression, where both induced the expression.

### 3.6. Impact of p53 Deletion and KRAS-G12D Mutation on Ferroptosis

To determine the relative importance of p53 deletion and the oncogenic mutation in KRAS in the protection of colon cancer cells from ferroptosis, we monitored ferroptosis in all four cell lines using a fluorescence assay that was based on cellular levels of ferroptosis-inducing lipid radicals. The HCT-116 parent cell line showed evidence of robust ferroptosis, and the deletion of p53 in this cell line almost completely protected the cells from this form of cell death (Figure 5). In contrast, the SW48 parent cell line possessed only a modest activity of ferroptosis, and more importantly, the G12D mutation in KRAS did not have any effect on this process (Figure 5). These differential effects of p53 deletion and KRAS mutation on ferroptosis occurred despite the fact that both induced SLC7A11 expression and function. This suggests that p53 loss is the predominant factor in the protection of colon cancer cells from ferroptosis, with little/no participation by the oncogenic mutation in KRAS in the process.

### 3.7. Influence of p53 Deletion and KRAS-G12D Mutation on Various Factors That Are Relevant to Ferroptosis

Ferroptosis is a non-apoptotic, non-necroptotic, and caspase-independent cell death caused the excessive oxidation of double bonds in polyunsaturated fatty acids in membrane-associated phospholipids. This involves reactive oxygen species (ROS) such as H_2_O_2_, and the process generates lipid peroxide radicals and malondialdehyde (MDA). The culprit in the whole process is excessive labile iron in the Fe^2+^ form that converts H_2_O_2_ into hydroxyl radicals, which initiate oxidation of the membrane lipids. Glutathione protects against ferroptosis because of its ability to remove H_2_O_2_ and lipid peroxides. To assess the role of these factors in ferroptosis in the two pairs of isogenic colon cancer cell lines in the present study, we quantified most of these parameters. The iron assay employed in our study measured only Fe^2+^, which is involved in the Fenton reaction. Deletion of p53 in the HCT-116 cells increased the cellular levels of Fe^2+^, whereas the presence of the oncogenic mutation G12D in the SW48 cells decreased the levels (Figure 6). The ROS levels were lower in the SW48 cells than in the HCT-116 cells, but neither the loss of p53 nor the presence of the G12D mutation in KRAS had any significant effect on the ROS levels in the respective cell lines (Figure 7). Even though the levels of glutathione (GSH) remained the same in the control and p53-null HCT-116 cells, the MDA levels, an indicator of ferroptosis, were lower in the *p53*-null cells (Figure 7). In contrast, the GSH levels and MDA levels were both increased in the SW48 cells harboring the G12D mutation in KRAS compared to the parent SW48 cells (Figure 7). Obviously, these parameters do not directly correlate with the ferroptosis status of the four cell lines, most likely because of the intricate back-and-forth relationship between oxidative stress and GSH, where oxidative stress induces GSH synthesis, whereas an increase in GSH levels suppresses oxidative stress. The same is true with Fe^2+^ and p53 because increased iron/heme promotes the degradation of p53, whereas the loss of p53 increases iron levels. However, the ferroptosis status is the final outcome that is important in understanding the role of p53 loss and oncogenic mutations in KRAS in cancer cells.

### 3.8. Effects of Niclosamide on Intracellular pH and the Transport Function of SLC38A5 and SLC7A11

Niclosamide is known to have multiple effects in cancer cells, influencing several signaling pathways [57,58,59]. We focused in the present study on the effects of this drug on intracellular pH and on the transport activities of SLC38A5 and SLC7A11 because of their direct connection to macropinocytosis (intracellular pH and SLC38A5) and ferroptosis (SLC7A11). Our recent studies with breast cancer cell lines have demonstrated that niclosamide causes intracellular acidification and also inhibits SLC38A5, both leading to the suppression of macropinocytosis [31]. Niclosamide is also a potent inhibitor of SLC7A11 in breast cancer cell lines, consequently leading to the induction of ferroptosis [30]. The effect on intracellular pH is related to the function of niclosamide as a protonophore (i.e., a H^+^ channel) [60,61], whereas the inhibitory effect of the drug on SLC38A5 and SLC7A11 is the result of its direct interaction with the two transporters. In the present study, we tested if niclosamide had similar effects in colon cancer cell lines. In all four cell lines examined in the present study, niclosamide (2.5 μM) caused a decrease in intracellular pH, leading to cellular acidification (Figure 8). The rate of acidification varied among the four cell lines; it was the highest in the HCT-116 cells with wild-type p53 and the lowest in the SW48 cells with the oncogenic mutation G12D in KRAS. Another noticeable thing was the influence of p53 loss in the HCT-116 cells and KRAS-G12D mutation in the SW48 cells. Both these changes are oncogenic, and these changes significantly decreased the rate of acidification by niclosamide. This suggests that genetic changes related to p53 and KRAS that promote cancer are associated with a significant impact on the control of intracellular pH by effectively protecting the cells from acidification.

We also monitored the effect of niclosamide on the transport activities of SLC38A5 (Li^+^-coupled serine uptake in the SW48 cells with KRAS-G12D) and SLC7A11 (Na^+^-independent glutamate uptake in the *p53*-null HCT-116 cells) (Figure 8). There was no specific reason to select these two cell lines except that p53 loss and KRAS-G12D mutation represented oncogenic changes that are relevant to promotion of cancer growth and that these cell lines showed the greatest activity for the respective transporters. We found that niclosamide had a robust inhibitory effect on the transport functions of SLC38A5 as well as SLC7A11. Even at concentrations of less than 0.5 μM, the inhibition in both transporters was 50% or greater.

### 3.9. Inhibition of Macropinocytosis by Niclosamide

Based on our previous studies with breast cancer cells [31,32], the ability of niclosamide to induce intracellular acidification and inhibit SLC38A5 is expected to suppress macropinocytosis. We tested this by monitoring macropinocytosis-mediated cellular entry of TMR-dextran in all four colon cancer cell lines with a 30-min preincubation with and without niclosamide (5 μM). Irrespective of the presence or absence of oncogenic changes in p53 and KRAS, niclosamide showed a dramatic suppressive effect on macropinocytosis (Figure 9). The inhibition was 70% or greater in all four cell lines.

### 3.10. Induction of Ferroptosis by Niclosamide

The marked inhibitory effect of niclosamide on the transport activity of SLC7A11 is expected to promote ferroptosis. We tested this by monitoring the cellular levels of lipid radicals with a fluorescence probe following a 30-min preincubation with and without niclosamide (5 μM). The data from the isogenic HCT-116 cells with and without p53 are given in Figure 10. In the parent cell line, which harbored wild-type p53, ferroptosis was robust even under basal conditions without exposure to niclosamide, and niclosamide did not have any effect on ferroptosis in this cell line. In contrast, ferroptosis was almost undetectable in the HCT-116 cells in which p53 was deleted. This was seen in the earlier experiment described in Figure 5. But, in this cell line, niclosamide was able to induce ferroptosis to a marked extent in a dose-dependent manner. A significant induction of ferroptosis was noticeable even at 0.25 μM niclosamide, and the maximal effect was seen with less than 5 μM niclosamide. In the SW48 cells, there was a modest activity of ferroptosis under basal conditions without exposure to niclosamide, which was not affected by the presence or absence of the G12D mutation in KRAS, as seen in an earlier experiment (Figure 5). However, exposure to niclosamide markedly induced ferroptosis in this cell line irrespective of the presence or absence of the oncogenic mutation G12D in KRAS, and the effect was dose-dependent (Figure 11). The potentiating effect of niclosamide on ferroptosis was noticeable even at 0.25 μM, and the maximal effect was observed at niclosamide concentrations of less than 5 μM. With the *p53*-null HCT-116 cells, we confirmed that the process induced by niclosamide in increasing the fluorescent signals for lipid radicals was indeed ferroptosis by demonstrating the blockade of the process by ferrostatin (Figure 12), a radical-trapping antioxidant and a selective blocker of ferroptosis [62,63]. We were also able to confirm this with the SW48 cells with and without the KRAS G12D mutation (Figure 12).

### 3.11. Effects of Niclosamide on Cellular Levels of ROS, GSH, and MDA

Since we observed a profound ability of niclosamide to induce ferroptosis in colon cancer cells, we monitored the influence of this drug on the cellular levels of reactive oxygen species (ROS), glutathione (GSH), and malondialdehyde (MDA), all of which are directly related to the process of ferroptosis. Cells were treated with 2.5 μM niclosamide for 24 h, and then the cells were lysed and used for measurements of ROS, GSH, and MDA. The results showed that niclosamide increased ROS levels and decreased GSH levels, which was accompanied with an increase in MDA levels; these findings were consistent in all four colon cancer cell lines (HCT-116 cells with and without p53 and SW48 cells with and without the G12D mutation in KRAS) (Figure 13).

### 3.12. Effects of Niclosamide on Cellular Levels of Ferritin Heavy Chain and Glutathione Peroxidase

We also monitored the impact of niclosamide treatment (2.5 μM; 24 h treatment) on the protein levels for the ferritin heavy chain FTH1 as well as for the antioxidant enzyme glutathione peroxidase (GPX). Niclosamide decreased the levels of FTH1 in the HCT-116 and SW48 cells independent of the presence or absence of p53 or the KRAS-G12D mutation (Figure 14). For GPX4, the decrease was evident only in the SW48 cells with and without the KRAS-G12D mutation and in the *p53*-null HCT-116 cells. There was no difference in the parent HCT-116 cells.

### 3.13. Effects of Niclosamide on Cell Viability/Proliferation and Colony Formation

The experiments detailed in the previous sections of this study have shown that niclosamide suppresses macropinocytosis, an effective nutrient-acquiring mechanism, and at the same induces ferroptosis, an iron-dependent cell death process. Therefore, these effects are expected to have a significant detrimental effect on the cell viability, cell proliferation, and colony-formation ability of colon cancer cells. We addressed this issue by examining the influence of niclosamide on these processes in all four cell lines. Niclosamide markedly inhibited cell viability and cell proliferation in a dose-dependent and exposure-time-dependent manner, as monitored by an MTT assay (Figure 15). Even at niclosamide concentrations as low as 2 μM, the inhibitory effect on cell viability and proliferation was 50% or greater. With a 72-h treatment, the inhibition reached >75%. Interestingly, the inhibition appeared to be more potent in the cells with oncogenic changes in p53 and KRAS. The effects were similar when the colony-forming ability of these cells was examined (Figure 16). The inhibition was significant even at concentrations of niclosamide as low as 0.5 μM; in three out of the four cell lines, the inhibition was 75% or greater at 1 μM niclosamide.

## 4. Discussion

Recently, we published reports on macropinocytosis and ferroptosis in the triple-negative breast cancer cell lines MB231 and TXBR-100 and on the effects of the anti-helminthic drug niclosamide on these two processes [31,32,45]. In these cell lines, niclosamide caused intracellular acidification and had opposing effects on macropinocytosis (suppression) and ferroptosis (induction). However, we did not examine the influence of p53 and KRAS on macropinocytosis and ferroptosis in these cell lines. Even though mutant p53 is common in breast cancer, mutations in KRAS are not frequent (<5%). Among the breast cancer cell lines, only MB231 possesses mutant p53 and also mutant KRAS (G13D) [64,65,66]. The mutational status of p53 and KRAS is not known for TXBR-100, a patient-derived breast cancer cell line that we used in these previous studies. As such, the previously published studies did not focus on the differential influence of p53 and KRAS mutations on macropinocytosis and ferroptosis in breast cancer cells. We also did not know if the actions of niclosamide as a suppressor of macropinocytosis and an inducer of ferroptosis that we observed in the breast cancer cells were dependent on the mutational status of p53 and/or KRAS. The present study represents the first on this topic, but it was conducted with colon cancer cells where isogenic cell lines were available with and without p53 and KRAS mutation. Moreover, such studies are more relevant to colon cancer, where KRAS mutations are much more prevalent than in breast cancer cells (30–50% versus <5%).

Glutamine is a critical amino acid for cancer cell growth and proliferation based on its connection to mTOR activation, nucleotide and protein synthesis, ATP production, lipid synthesis, and antioxidant machinery [67]. Among the various glutamine transporters in mammalian cells [20], SLC38A5 is unique because its transport function is associated with intracellular alkalinization [68] and one-carbon metabolism [69]. This transporter has come to prominence in recent years because of the recognition that it is upregulated in breast cancer [45,70] and pancreatic cancer [71,72] and is coupled to chemoresistance [70,72,73]. The role of SLC38A5 in colon cancer has not yet been investigated even though it is expressed in the intestinal tract [68].

The finding that SLC38A5 is involved in macropinocytosis is a recent observation [45]. We hypothesized this involvement primarily based on the functional feature of the transporter as an amino-acid-dependent Na^+^/H^+^ exchanger and the known fact that Na^+^/H^+^ exchanger is an inducer of macropinocytosis. The role of Na^+^/H^+^ exchanger in macropinocytosis is also the basis for the widespread use of ethylisopropylamiloride and dimethylamiloride, potent inhibitors of the exchanger, as specific blockers of macropinocytosis [74].

The salient findings in the present study with regard to macropinocytosis in colon cancer cells can be summarized as follows: (i) the oncogenic mutation G12D in KRAS is the driver of macropinocytosis with apparently no involvement of p53; (ii) the oncogenic changes in p53 (loss of function) and KRAS (activating mutation) elicit opposite effects on SLC38A5 expression, and the induction of the transporter by KRAS-G12D mutation is at least partly responsible for the increased activity of macropinocytosis seen in SW48 cells with the mutation; (iii) the increase in SLC38A5 expression elicited by the KRAS-G12D mutation is mediated by the transcriptional activity of the oncogene Myc; (iv) SLC38A5 is also a target for p53, but this transcription factor functions as an inducer of SLC38A5 expression in contrast to its action as a suppressor of SLC7A11 expression; (v) niclosamide is a potent suppressor of macropinocytosis in colon cancer cells irrespective of whether or not the cells harbor oncogenic changes in p53 and KRAS; and (vi) the ability of niclosamide to cause intracellular acidification and to inhibit SLC38A5 is at least partly responsible for this effect.

Ferroptosis is being increasingly recognized as one of the most important vulnerabilities and an Achilles’ heel in cancer cells because these cells have to navigate between their obligate need to accumulate iron to support their growth and proliferation and at the same time the challenge to protect themselves from iron-induced cell death [13,14,75,76]. The tumor suppressor p53 and the amino acid transporter SLC7A11 are at the center of these two opposing phenomena. When cancer cells accumulate iron, heme levels increase. p53 is a heme-binding protein, and when heme binds to p53, the complex gets degraded in proteasomes [23]. As a consequence, excess iron in cancer cells leads to p53 depletion. SLC7A11 is the most important protector of cancer cells from iron-induced ferroptosis because of its ability to increase cellular levels of glutathione. The expression of this transporter is suppressed by p53 [23]. As such, p53 loss induced by excess iron/heme results in an increased expression of SLC7A11 in cancer cells, thus protecting the cells from ferroptosis. This phenomenon also occurs in hemochromatosis, a genetic disorder associated with iron overload, where iron accumulation leads to increased heme and consequent p53 loss and also increased susceptibility to colon cancer [77]. The present study provides confirmation of the functional interaction among p53, SLC7A11, and ferroptosis. The deletion of p53 in HCT-116 cells potentiates the expression and function SLC7A11, and as a result, ferroptosis becomes almost undetectable in these cells. Another important finding from the present study is that it is p53, not KRAS, that plays the key role in the control of ferroptosis. However, we found that the oncogenic mutation G12D in KRAS does induce the expression of SLC7A11 in SW48 cells even in the presence of p53. These findings suggest that KRAS activation in cancer cells has the potential to provide protection against ferroptosis under conditions where p53 is still functional. It is interesting to note the similarity in the transcriptional activation of SLC7A11 between the oncogenic changes in p53 (loss of function) and KRAS (activating mutation), both leading to the induction of SLC7A11 expression. This is in contrast to their effects on SLC38A5 expression, where they play opposing roles.

In addition to providing important new information on the opposing roles of oncogenic changes in p53 and KRAS in colon cancer cells not only in terms of macropinocytosis and ferroptosis but also in terms of the control of SLC38A5 and SLC7A11 expression, the present study highlights the therapeutic potential of the FDA-approved anti-helminthic drug niclosamide. The findings that the drug elicits opposing effects on macropinocytosis and ferroptosis in colon cancer cells irrespective of the presence or absence of oncogenic changes in p53 and KRAS are the salient features of the present study. The drug blocks macropinocytosis, thus interfering with an important route of nutrient acquisition in cancer cells; at the same time, it also induces ferroptosis, a key vulnerability in cancer cells. These effects are at least partly due to the ability of the drug to inhibit the transport function of SLC38A5 (relevant to macropinocytosis) and SLC7A11 (relevant to ferroptosis). It is also important to point out here that the potency of niclosamide as an inhibitor of SLC7A11 (*IC*_50_, <0.25 μM) is greater than most of the inhibitors, including erastin, reported thus far in the literature [78]. We found a similar *IC*_50_ value in triple-negative breast cancer cells [32]. Even though derivatives of erastin have been developed recently with a much greater potency to inhibit SLC7A11 (*IC*_50_ values in the low-nanomolar range) [79], niclosamide may have an advantage in terms of therapeutic use for cancer therapy because of its feature as an already FDA-approved drug that has been in use in humans for several decades. The often-pointed-out disadvantage of niclosamide as an anticancer agent is its poor oral bioavailability [80,81]. This may be a valid argument for the use of the drug to treat cancers of a non-colonic origin but not for its use to treat colon cancer. Niclosamide is active in the lumen of the intestinal tract when given for the treatment of helminth infections; therefore, the drug will have access to cancer cells in the colon to elicit its anticancer effects. With this rationale, we conclude that the results of the present study underscore the therapeutic potential of niclosamide in the treatment of colorectal cancer. Even for other cancers of a non-colonic origin, there could be effective strategies such as nano-formulations or chemical modifications to improve the oral bioavailability of niclosamide to harness its efficacy as an anticancer drug.

## 5. Conclusions

There are two important conclusions that can be gleaned from the present study: (i) macropinocytosis is a pro-growth process in cancer that is primarily promoted by activating mutations in KRAS in colon cancer cells, whereas ferroptosis is an anti-growth process in cancer that is primarily blocked by the loss of p53 in colon cancer cells; and (ii) the anti-helminthic niclosamide is a potent blocker of macropinocytosis and a potent inducer of ferroptosis in colon cancer cells, both effects having a marked anticancer impact irrespective of the presence or absence of oncogenic changes in p53 and KRAS.

## Figures and Tables

**Figure 1 cells-13-00951-f001:**
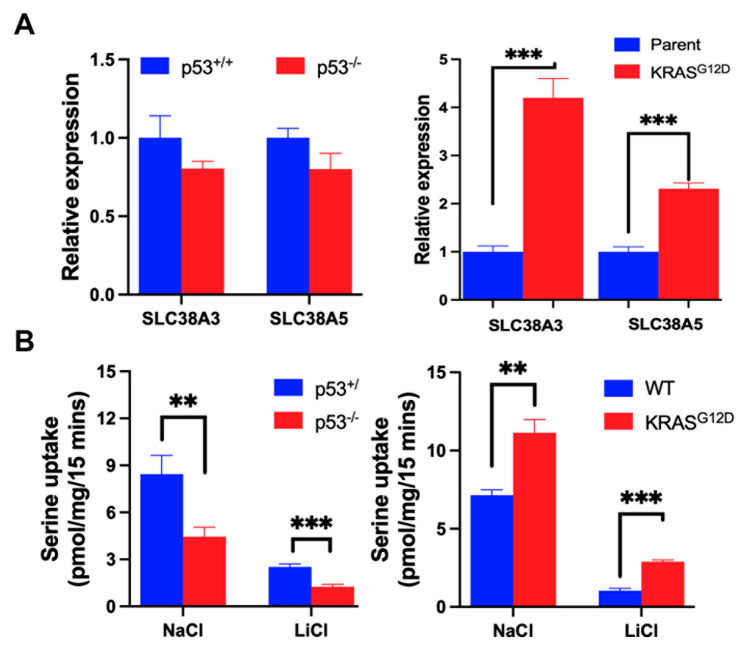
Expression and function of SLC38A3/SLC38A5 in the two pairs of isogenic colon cancer cell lines with and without p53 (HCT-116) and with and without G12D mutation in KRAS (SW48). Relative expression of mRNA levels as assessed by qRT-PCR (**A**). Transport function as assessed by serine uptake (**B**). Data are means ± S.E. for three independent experiments. **, *p* < 0.01; ***, *p* < 0.001. When not specified, the difference was not statistically significant.

**Figure 2 cells-13-00951-f002:**
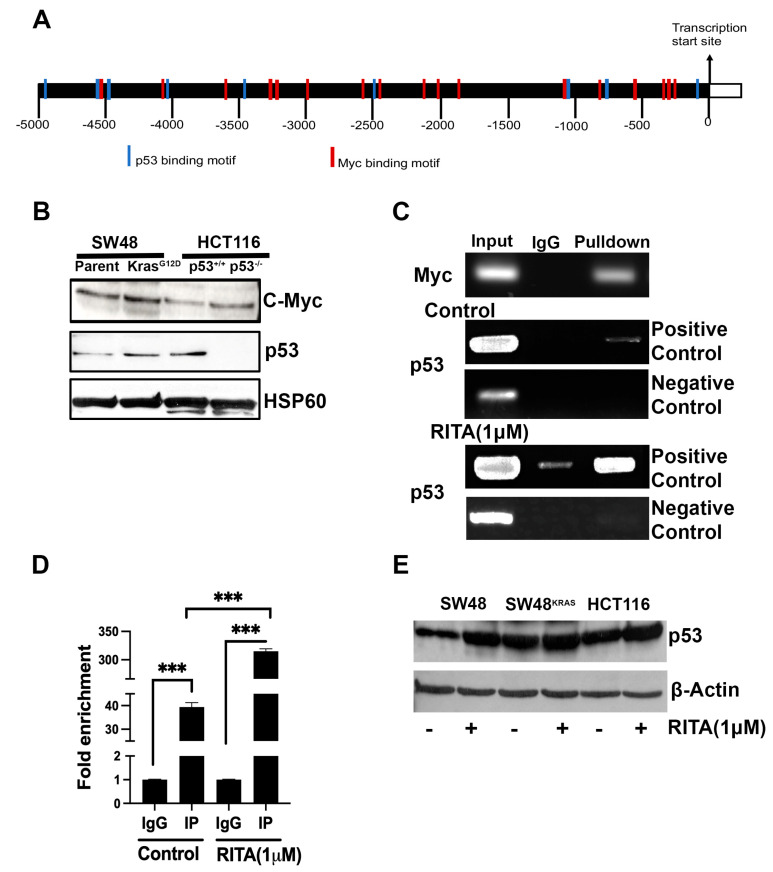
Evidence for SLC38A5 as a transcriptional target for p53 and Myc. Locations for theoretical binding sites in human SLC38A5 gene promoter for p53 and Myc (**A**). Protein levels for p53 and Myc in the isogenic cell lines SW48 with and without KRAS-G12D mutation and HCT-116 with and without p53 (**B**). SW48 parent cells with or without treatment in the presence of the p53 activator RITA (1 μM; 16 h treatment) were used for ChIP assay to provide evidence for the binding of p53 and Myc to the SLC38A5 gene promoter (**C**,**D**). The ability of RITA to increase the steady-state levels of p53 was confirmed by Western blotting in all three colon cancer cell lines that expressed wild-type p53 (**E**). ***, *p* < 0.001.

**Figure 3 cells-13-00951-f003:**
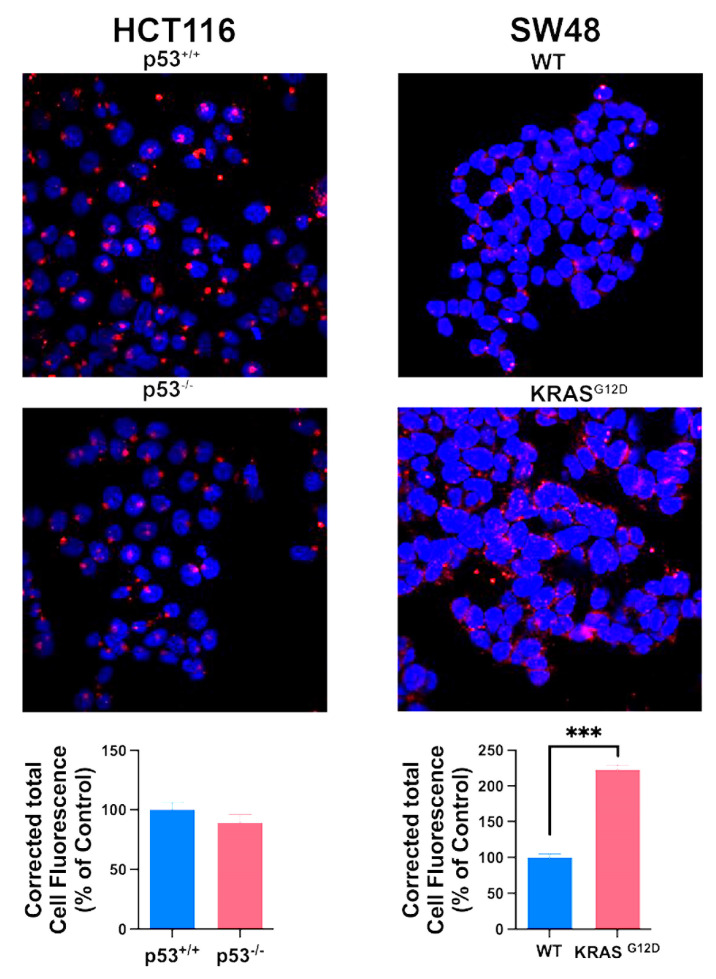
Effect of p53 deletion and oncogenic KRAS mutation G12D on macropinocytosis. Cellular uptake of TMR-dextran was used to monitor macropinocytosis activity. The fluorescence signals were quantified as CTCF (corrected total cell fluorescence) for all four cell lines. Data are means ± S.E. ***, *p* < 0.001. When not specified, the difference was not statistically significant.

**Figure 4 cells-13-00951-f004:**
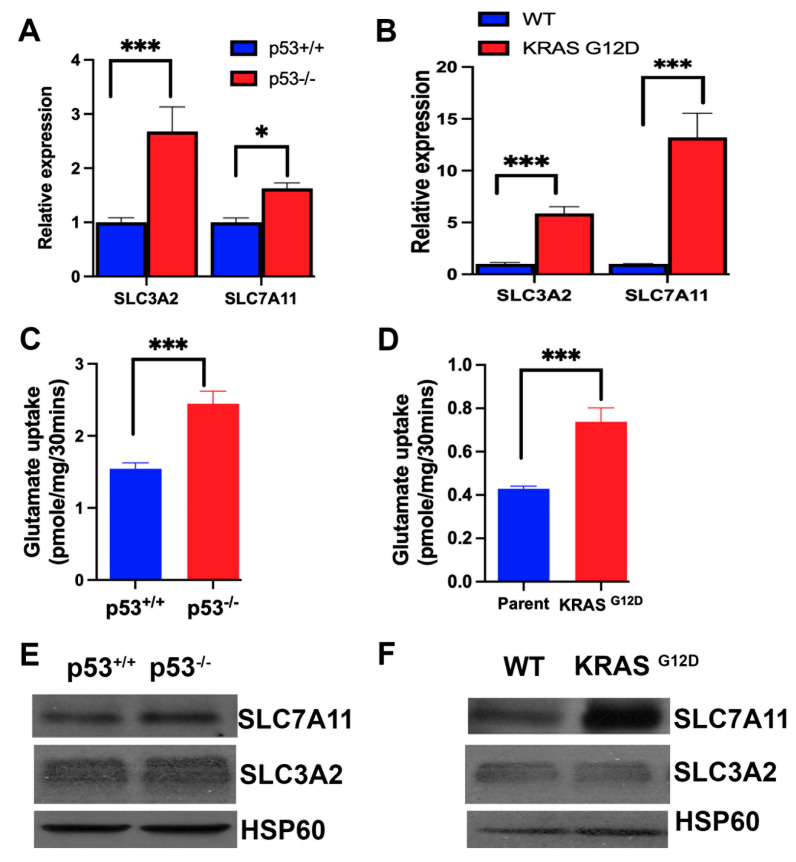
Expression and function of SLC7A11 and its chaperone SLC3A2 in the two pairs of isogenic colon cancer cell lines with and without p53 (HCT-116) and with and without G12D mutation in KRAS (SW48). Relative expression of mRNA levels as assessed by qRT-PCR (**A**,**B**). Transport function as assessed by glutamate uptake (**C**,**D**). Data are means ± S.E. for three independent experiments. *, *p* < 0.05; ***, *p* < 0.001. Protein levels for SLC7A11 and SLC3A2 in the four cell lines (**E**,**F**).

**Figure 5 cells-13-00951-f005:**
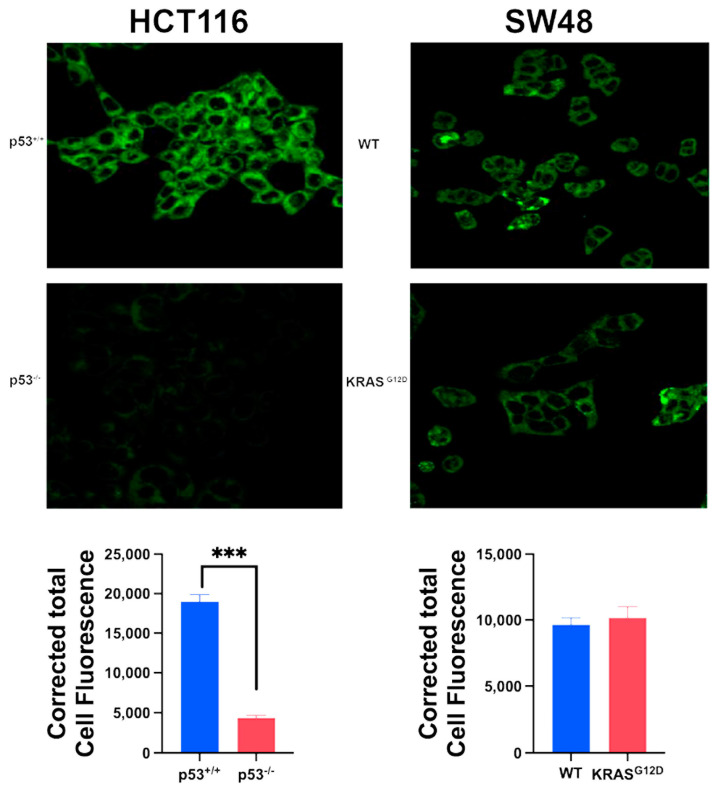
Basal ferroptosis activity in the two pairs of isogenic colon cancer cells with and without p53 (HCT-116) and with and without G12D mutation in KRAS (SW48) as assessed by fluorescence detection of lipid radicals. Quantifications of the fluorescence signals are also given (means ± S.E. for three independent experiments). ***, *p* < 0.001. When not specified, the difference was not significant.

**Figure 6 cells-13-00951-f006:**
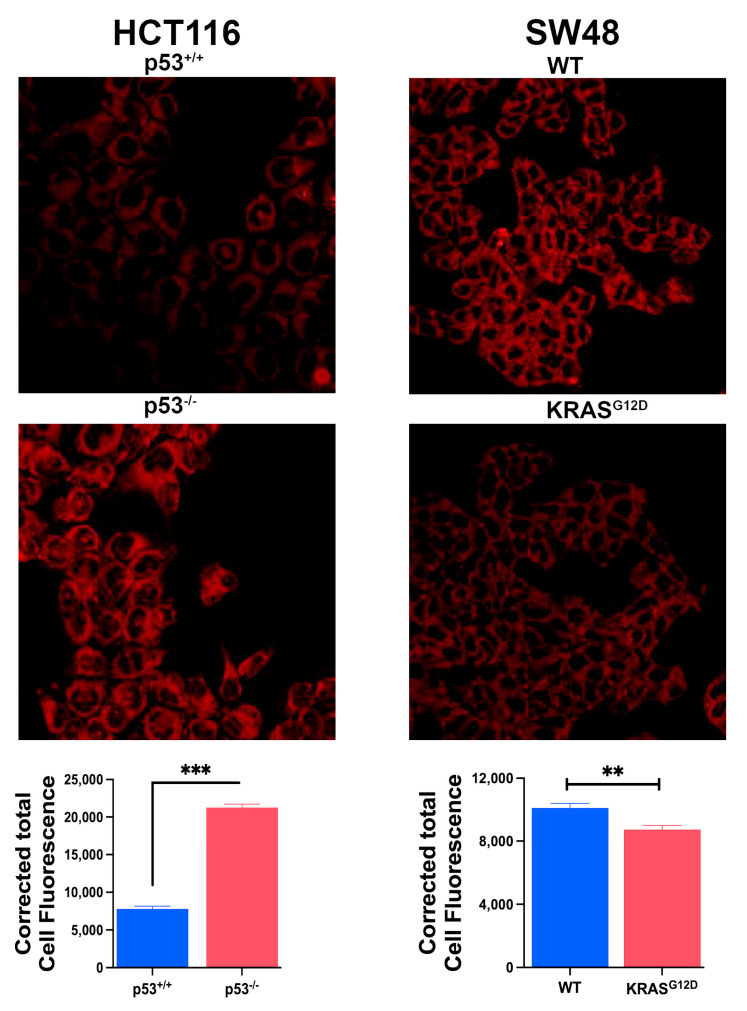
Iron levels in the two isogenic cell lines with and without p53 (HCT-116) and with and without G12D mutation in KRAS (SW48). The fluorescence signals were also quantified (data given as means ± S.E.). **, *p* < 0.01; ***, *p* < 0.001.

**Figure 7 cells-13-00951-f007:**
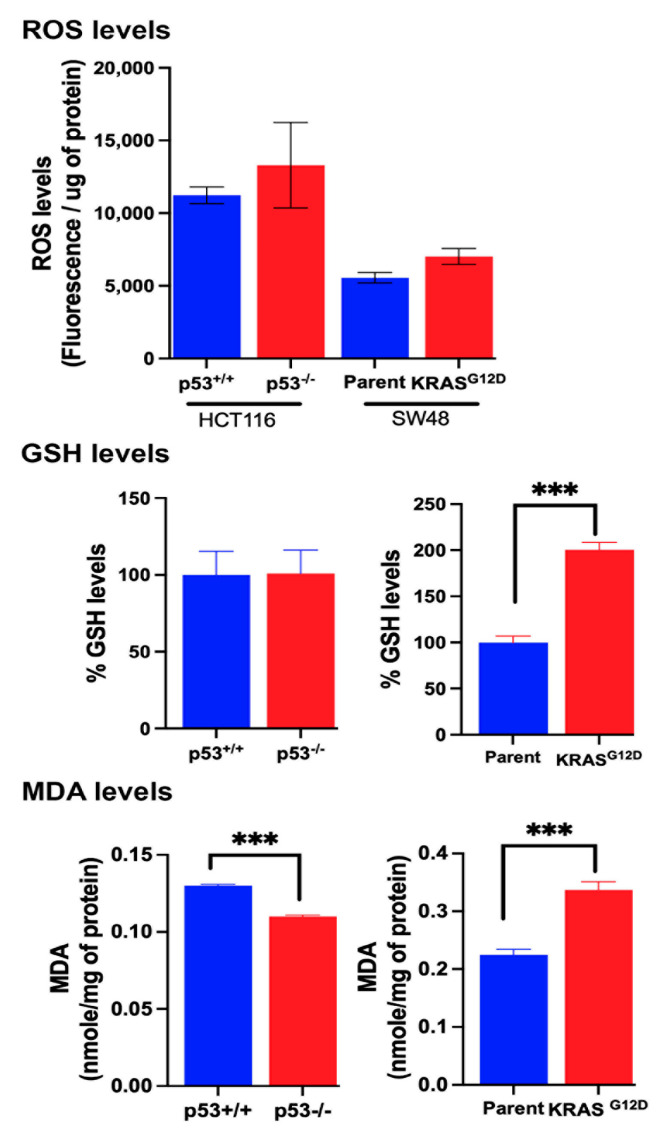
Basal levels of ROS, GSH, and MDA. Data (means ± S.E.) are from three independent experiments. ***, *p* < 0.001. When not specified, the differences were not statistically significant.

**Figure 8 cells-13-00951-f008:**
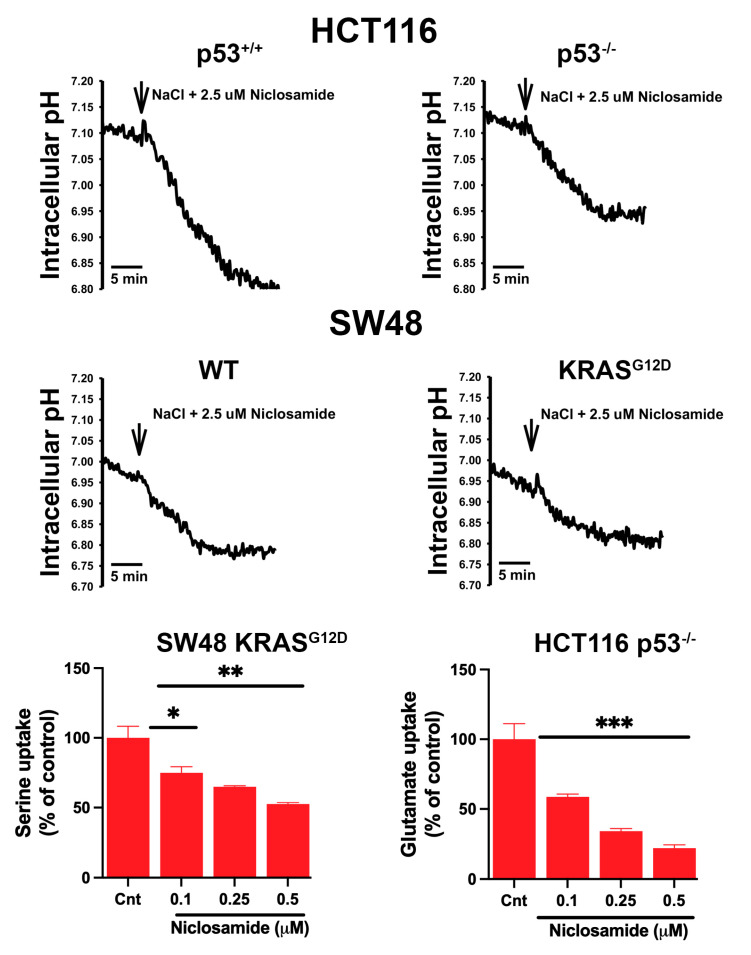
Effects of niclosamide on intracellular pH in all four cell lines and on the transport activity of SLC38A5 in SW48 cells with G12D mutation in KRAS and on the transport activity of SLC7A11 in p53-null HCT-116 cells. Data are given as means ± S.E. *, *p* < 0.05; **, *p* < 0.01; ***, *p* < 0.001.

**Figure 9 cells-13-00951-f009:**
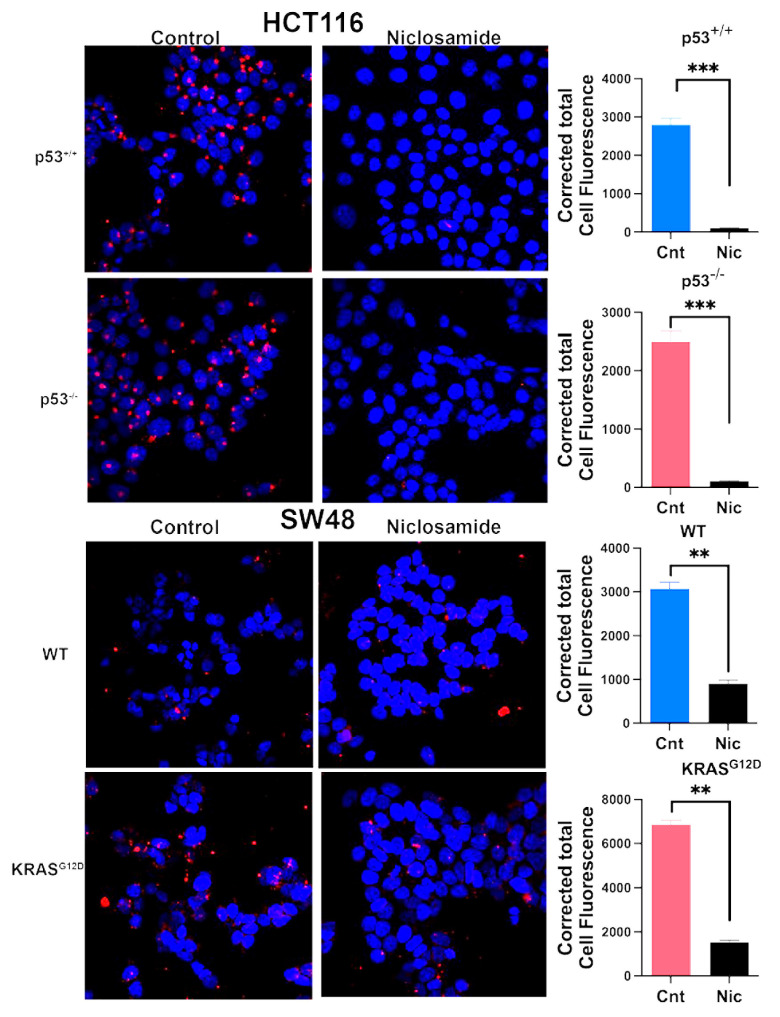
Effects of niclosamide (5 µM) on macropinocytosis in the two pairs of isogenic cell lines. Immunofluorescence images as well as quantification of the fluorescence signals are given. Data are given as means ± S.E. **, *p* < 0.01; ***, *p* < 0.001; Nic, niclosamide.

**Figure 10 cells-13-00951-f010:**
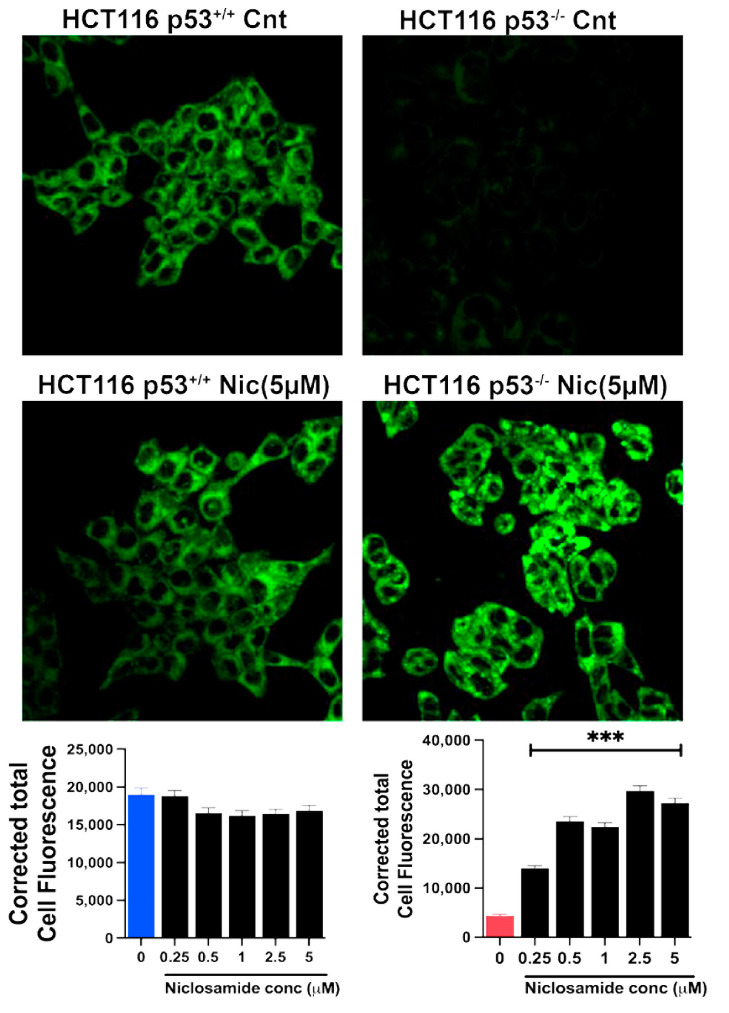
Induction of ferroptosis by niclosamide (5 µM) in HCT116 cells with and without p53. Quantification of the fluorescence signals (mean ± S.E.) are also given. ***, *p* < 0.001. When not specified, the differences were not statistically significant.

**Figure 11 cells-13-00951-f011:**
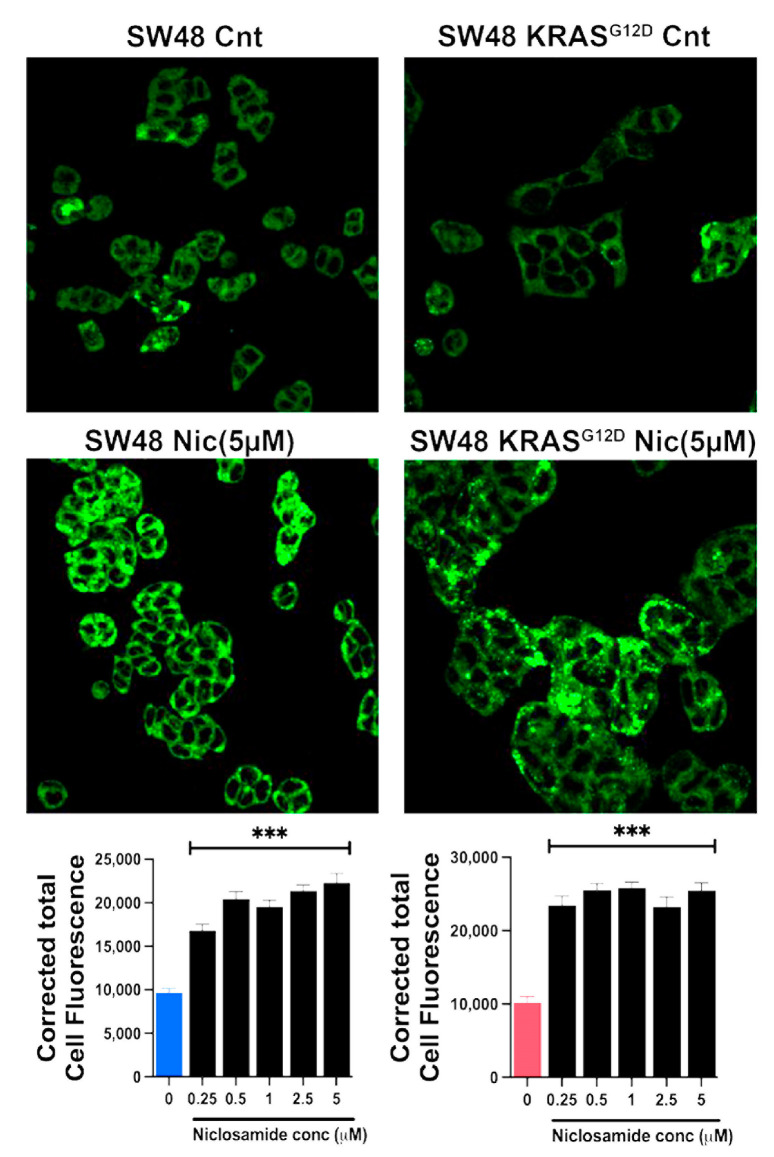
Induction of ferroptosis by niclosamide (5 µM) in SW48 cells with and without G12D mutation in KRAS. Quantification of the fluorescence signals (mean ± S.E.) are given. ***, *p* < 0.001.

**Figure 12 cells-13-00951-f012:**
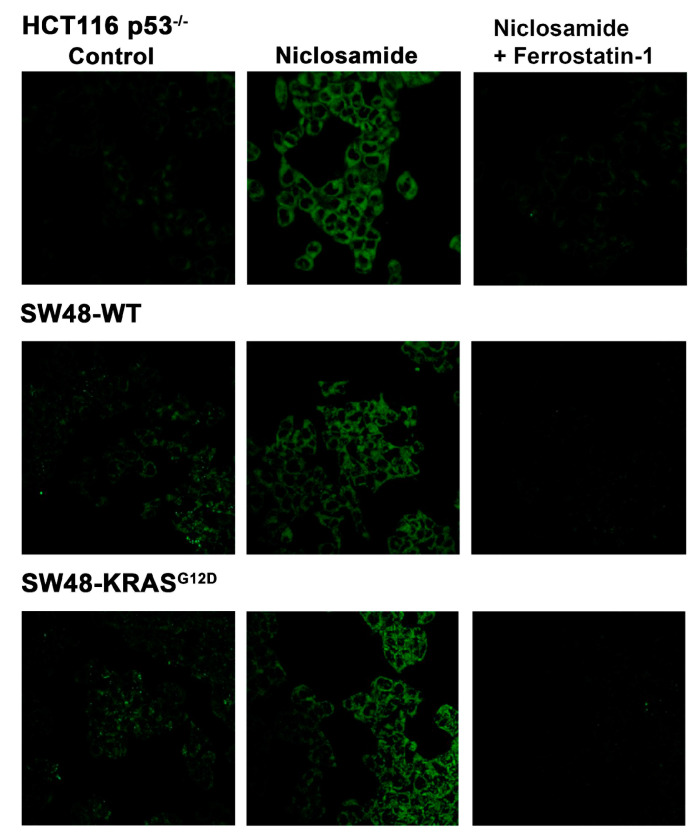
Blockade of niclosamide (5 µM)-induced ferroptosis by ferrostatin (10 µM) in *p53*-null HCT116 cells and in SW48 cells with and without the KRAS G12D mutation.

**Figure 13 cells-13-00951-f013:**
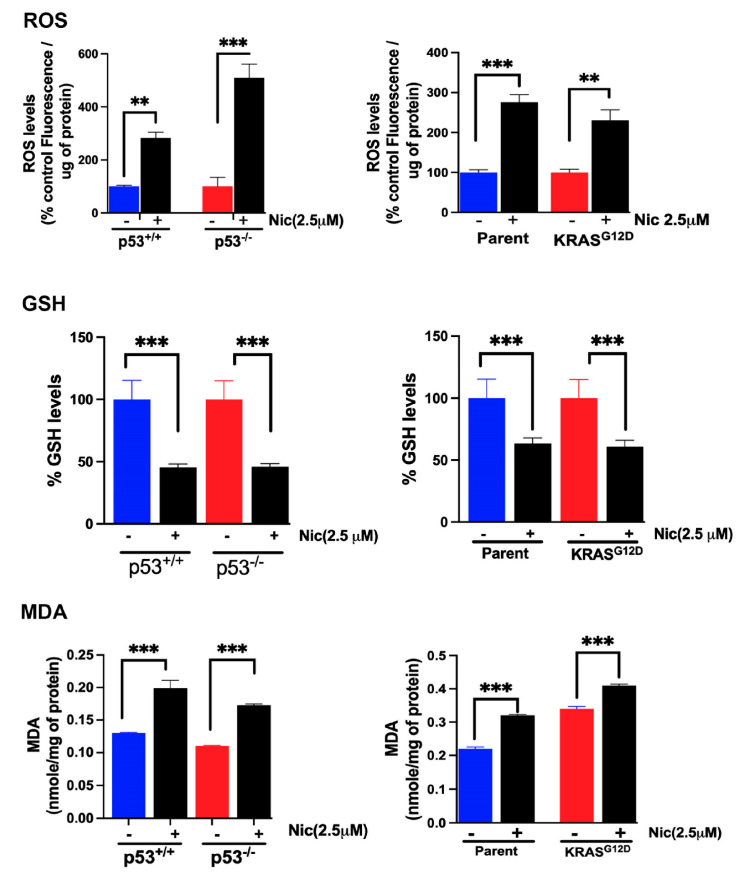
Effects of niclosamide on cellular levels of ROS, GSH, and MDA. The cells were treated with niclosamide (2.5 μM) for 24 h. Data are given as means ± S.E. **, *p* < 0.01; ***, *p* < 0.001.

**Figure 14 cells-13-00951-f014:**
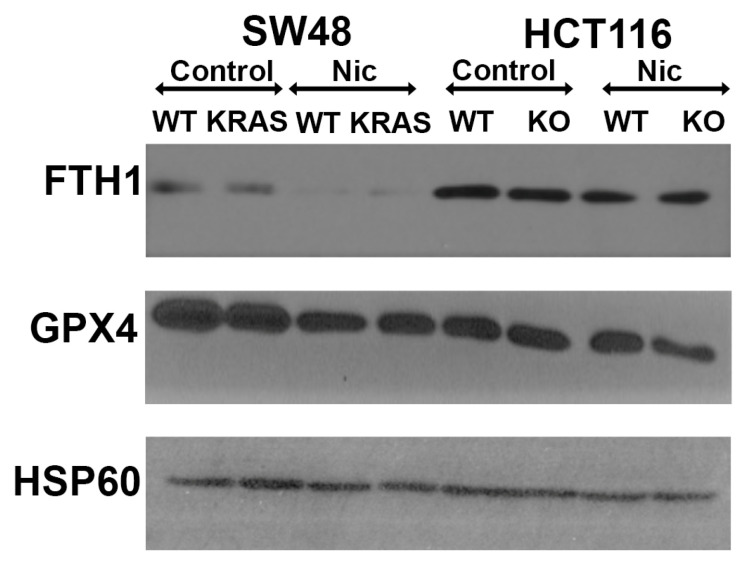
Effects of niclosamide treatment (2.5 μM; 24 h) on FTH1 and GPX4 protein levels.

**Figure 15 cells-13-00951-f015:**
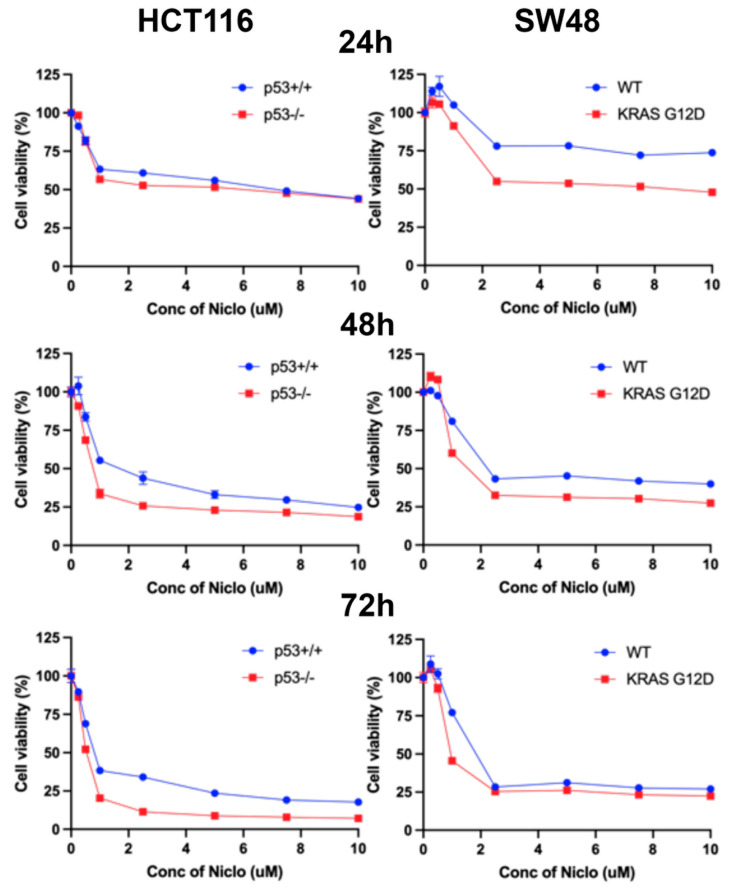
Inhibition of cell viability and proliferation by niclosamide as monitored by MTT assay in all four cell lines (HCT-116 cells with and without p53 and SW48 cells with and without KRAS-G12D mutation). Cell viability was calculated as percentage of control cells without treatment with niclosamide. Data (means ± S.E.) are from three independent experiments.

**Figure 16 cells-13-00951-f016:**
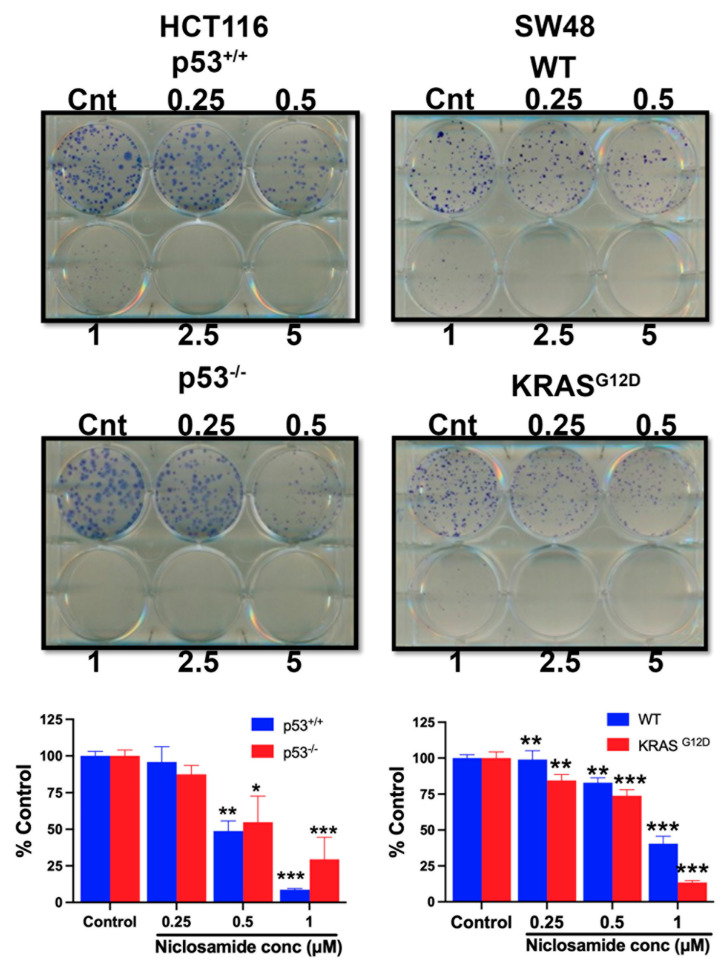
Effects of niclosamide on colony-forming ability in HCT-116 cells with and without p53 and in SW48 cells with and without KRAS-G12D mutation. Data (means ± S.E.) are from three independent experiments. *, *p* < 0.05; **, *p* < 0.01; ***, *p* < 0.001. When not specified, the differences were not statistically significant.

**Table 1 cells-13-00951-t001:** Genetic background of the four human colon cancer cell lines used in the study.

Gene/Cell Line	HCT-116	HCT-116/p53 KO	SW48	SW48/KRAS Mutant
APC	Wild-type	Wild-type	Wild-type	Wild-type
KRAS	pGly13Asp (G13D)	pGly13Asp (G13D)	Wild-type	pGly12Asp (G12D)
TP53 (p53)	Wild-type	Deleted	Wild-type	Wild-type

## Data Availability

The authors confirm that the data supporting the findings of this study are given in their entirety within this manuscript and its Appendix A.

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
