# Peer review of "Impact of Oncogenic Changes in p53 and KRAS on Macropinocytosis and Ferroptosis in Colon Cancer Cells and Anticancer Efficacy of Niclosamide with Differential Effects on These Two Processes"

_cells, 2024, doi:10.3390/cells13110951_

Round 1
Reviewer 1 Report
Comments and Suggestions for Authors
The results of the present study highlight the therapeutic potential of niclosamide in the treatment of colorectal cancer. Even for other cancers not originating in the colon, there could be effective strategies such as nanoformulations or chemical modifications to improve the oral bioavailability of niclosamide and harness its efficacy as an anticancer drug.
In summary, there are two important conclusions that can be drawn from the present study: (i) Macropinocytosis is a pro-growth process in cancer primarily promoted by activating mutations in KRAS in colon cancer cells, while ferroptosis is an anti-growth process in cancer primarily blocked by loss of p53 in colon cancer cells; (ii) Niclosamide, an anti-helminthic drug, is a potent blocker of macropinocytosis and a potent inducer of ferroptosis in colon cancer cells, both effects having a marked anticancer impact, irrespective of the presence or absence of oncogenic changes in p53 and KRAS.
This is a highly relevant research, and I find the approach, objectives, and results obtained to be very appropriate, with well-considered conclusions and a thoughtful discussion.
Author Response
This reviewer had no concerns ragarding the original version. Therefore, we did not change anything related to the critique by this reviewer.
We thank the reviewer for his/her positive comments.
Reviewer 2 Report
Comments and Suggestions for Authors
Nhi T. Nguyen and colleagues report on the potential impact of the loss of p53 or the expression of oncogenic KRAS in colon cancer-derived cell lines on the processes of micropinocytosis and ferroptosis.
The Authors are building the study based on previous research that was performed in other cancer cell lines of different tissue origin. Thus, this manuscript represents an extension of earlier findings with some novel mechanistic insights.
The manuscript is well written, methods are well documented, and figures are, for the most part, presenting convincing results.
However, one severe limitation of the work is the choice of two different pairs of cell lines to perform the various assays. Furthermore, in the case of p53, the choice of not deriving new isogenic p53-null clones using editing approaches represents to me a missed opportunity and one that reduces the potential impact of the observations made in the paper. For example, statements like the one on lines 467-468 on the effect of “p53 depletion” on ferroptosis are, in my view, partially misleading. If the deletion of p53 by CRISPR technology is complicated, perhaps an actual depletion experiment could be considered by transient transfection. In principle, this could also enable modulating p53 status in SW48 to explore possible crosstalk between KRAS G12D and p53 status.
Specific points
-The inspection of the SLC38A5 promoter for p53 binding sites (Figure 2) led to the identification of several putative hits. This is surprising given the lack of evidence for p53 binding to the SLC38A5 promoter or enhancer, based on many available ChIP-seq datasets (summarized, for example, in a database curated by Fischer’s lab). Figure 2 should be improved by providing more information about the actual positional weight matrices used for the prediction and by providing more quantitative ChIP data, including positive and negative controls. Also, it appears that the ChIP experiment was performed in cells that were not treated to activate/stabilize wild-type p53. This important control should also be included.
-Why is niclosamide treatment leading to the induction of ferroptosis in the HCT116 p53 null cell line while it has no impact (if at all. a slight reduction) in HCT116 p53 wild-type cells (Figures 10 and 11)? Ferroptosis is induced in SW48 cells that are also p53 wild type.
-Is Ferrostatin treatment blocking ferroptosis induced by niclosamide also in SW48 cells?
Author Response
We thank the reviewer for his/her constructive criticisms. We performed several experiments to address his/her concerns. We have now incorporated all the new data from these additional experiments. The revised version (marked copy) highlights all the changes made in the revision.
Below are the specifics of the changes made.
- The reviewer raised some concerns about the HCT-116 cells, questioning whether the p53-positive and p53-negative cell lines are truly isogenic. They are indeed isogenic cell lines, generated in Johns Hopkins University in the laboratory of Dr. Bert Vogelstein. We have now given the details of these cell lines in the Methods section where we describe the cell lines and also added two new references from Dr. Vogelstein's lab that detail the generation of the cell lines.
- The other concern the reviewer raised is to do with the p53-binding sites in SLC38A5 gene promoter. We have rechecked the website and confirmed the presence of the binding sites in the gene promoter for the transcription factor p53. In addition, the reviewer asked us to do an additional experiment using an activator of p53 in the ChIP assay to confirm the presence of p53 binding site. We have now done this extra experiment using the p53 activator RITA. The results of this experiment did confirm the increased binding of p53 to the promoter in the ChIP assay. The new data are now given in the revised Fig. 2. We also confirm the increased steady-state levels of p53 in RITA-treated cells. This new data are also given in revised Fig. 2.
- The reviewer asked us to do the experiment with ferrostatin in SW48 cells to confirm that the cell death process induced by niclosamide is indeed ferroptosis not only in HCT-116 cells but also in SW48 cells. We have done this experiment as requested by the reviewer. The data are now included in the revised Fig. 12.
With these additional experiments, we have addressed all the concerns raised by this reviewer. We sincerely hope that the reviewer is now satisfied with the revised version.
Round 2
Reviewer 2 Report
Comments and Suggestions for Authors
The Authors submitted a revised version of their study that contains two additional experiments suggested during the revision process. The ChIP data added in Figure 2 is not completely convincing -missing negative controls and nondisclosure of which binding site was interrogated by PCR among the many mapped on the sequence upstream TSS-. The stabilization of p53 by RITA does not appear to be substantial in the western blot; hence, the near 8-fold induction in occupancy is unexpected. Further, the Authors seem to have missed the point about the comment about HCT116 cells. Nevertheless, the revised version of the manuscript contains some improvements and the new data is consistent overall with the proposed model.
Comments on the Quality of English LanguageMinor editing of English language is advised
Author Response
We thank the reviewer for the comments. Apparently, we did miss the reviewer's point about the HCT-116 cells. We thought that he/she doubted the isogenic nature of the p53-positive and p53-null HCT-116 cells. That was why we provided the original references of the publications from Dr. Vogelstein's lab to confirm that these are indeed isogenic cell lines. Now we realize that what the reviewer wanted us to do was to manipulate p53 levels in HCT-116 cells (e.g., siRNA or shRNA in p53-positive cells; ectopic expression in p53-null cells). We apologize for this misunderstanding. However, we were not able to perform these experiments for the current revision because of the 5-day deadline for the submission of the revised manuscript. We sincerely hope that the data from the isogenic cell lines themselves are sufficient enough to demonstrate the role of p53 in the observed changes in macropinocytosis and ferroptosis as concluded in the manuscript.
The non-disclosure of the exact positions of the PCR primers used in the ChIP assay was not intentional; it was just an unfortunate oversight on our part. We have provided the necessary details about the primers in the revised version. With regard to the negative control, we did do these experiments but did not include the data because we felt that the inclusion of the data with the p53 stabilizer (RITA) was sufficient for confirmation of the binding of p53 to the promoter. We have now included the data for the negative control in the revised version (Fig. 2C).
We agree with the reviewer about the discrepancy in the changes in the protein levels of p53 and the degree of occupancy on the promoter with and without treatment with RITA. We repeated the qPCR for the ChIP assay to determine if there was any mistake in our earlier experiment, but we obtained the same data. One possible explanation could be a difference in IP efficiency depending on the p53 protein levels. Nonetheless, we hope that the reviewer would be satisfied that the data with RITA do provide supporting evidence confirming the binding of p53 to the SLC38A5 promoter.
We truly thank the reviewer for the critical evaluation of the manuscript that mandated the additional experiments with the resultant improvement in the quality of the final manuscript.